# Improving sepsis prediction in intensive care with *SepsisAI*: A clinical decision support system with a focus on minimizing false alarms

**Ankit Gupta**◉*, Ruchi Chauhan, Saravanan G, Ananth Shreekumar

Center for Innovation in Diagnostics, Siemens Healthcare Private Limited, Bangalore, India

\* gupta.ankit@siemens-healthineers.com

**Data Availability Statement:** The data that support the findings of this study are publicly available from physionet repository with the identifier(s) https://physionet.org/content/challenge-2019/1.0.0/.

## Abstract

Prediction of sepsis using machine-learning approaches has recently gained traction. However, the lack of translation of these algorithms into clinical routine remains a major issue. Existing early sepsis detection methods are either based on the older definition of sepsis or do not accurately detect sepsis leading to the high frequency of false-positive alarms. This results in a well-known issue of clinicians' "alarm fatigue", leading to decreased responsiveness and identification, ultimately resulting in delayed clinical intervention. Hence, there is a fundamental, unmet need for a clinical decision system capable of accurate and timely sepsis diagnosis, running at the point of need. In this work, SepsisAI–a deep-learning algorithm based on long short-term memory (LSTM) networks was developed to predict the early onset of hospital-acquired sepsis in real-time for patients admitted to the ICU. The models are trained and validated with data from the PhysioNet Challenge, consisting of 40,336 patient data files from two healthcare systems: Beth Israel Deaconess Medical Center and Emory University Hospital. In the short term, the algorithm tracks frequently measured vital signs, sparsely available lab parameters, demographic features, and certain derived features for making predictions. A real-time alert system, which monitors the trajectory of the predictions, is developed on top of the deep-learning framework to minimize false alarms. On a balanced test dataset, the model achieves an AUROC, AUPRC, sensitivity, and specificity of 0.95, 0.96, 88.19%, and 96.75%, respectively at the patient level. In terms of lookahead time, the model issues a warning at a median of 6 hours (IQR 6 to 20 hours) and raises an alert at a median of 4 hours (IQR 2 to 5 hours) ahead of sepsis onset. Most importantly, the model achieves a false-alarm ratio of 3.18% for alerts, which is significantly less than other sepsis alarm systems. Additionally, on a disease prevalence-based test set, the algorithm reported similar outcomes with AUROC and AUPRC of 0.94 and 0.87, respectively, with sensitivity, and specificity of 97.05%, and 96.75%, respectively. The proposed algorithm might serve as a clinical decision support system to assist clinicians in the accurate and timely diagnosis of sepsis. With exceptionally high specificity and low false-alarm rate, this algorithm also helps mitigate the well-known issue of clinician alert fatigue arising from currently proposed sepsis alarm systems. Consequently, the algorithm partially addresses

**Funding:** The author(s) received no specific funding for this work.

**Competing interests:** The authors have declared that no competing interests exist.

the challenges of successfully integrating machine-learning algorithms into routine clinical care.

## Author summary

Sepsis is a serious life-threatening complication that results from the body's exaggerated response to an infection, leading to organ dysfunction. Distinguishing sepsis from other inflammatory conditions is challenging, leading to significant morbidity, mortality, and healthcare costs. We've developed a deep-learning based approach—SepsisAI, leveraging Long Short-Term Memory networks. It enables real-time monitoring and prediction of hospital-acquired sepsis for ICU patients, using routine parameters. The algorithm, trained and validated using data from two healthcare systems, demonstrates impressive results. Achieving a high AUROC, AUPRC, sensitivity, and specificity, it predicts sepsis hours before sepsis onset. It also uses a warnings and alert system resulting in a notably low false-alarm ratio, hence addressing the prevalent issue of alert fatigue, marking a positive step towards integrating machine-learning into routine clinical care. Our goal is that this effort will ultimately enhance patient survival and yield positive outcomes in terms of antimicrobial stewardship for complex and diverse conditions, such as sepsis.

## Introduction

Sepsis is a life-threatening organ dysfunction caused by the dysregulated host response to an infection and is responsible for significant mortality, morbidity, and healthcare expenses [1,2]. Globally, more than 49 million people are affected by sepsis which accounts for 19.7% of all global deaths [3]. Since signs and symptoms of sepsis are often non-specific, distinguishing sepsis from other types of inflammatory conditions is challenging [4]. More importantly, heterogeneity in infection sources, immune responses, and pathophysiological changes among septic patients result in failure to establish a prompt and accurate diagnosis, leading to unacceptably high mortality rates [5]. The ongoing debates over sepsis definitions and clinical criteria, as evidenced by the recently proposed redefinitions of sepsis [6], underscore a fundamental difficulty in its identification and accurate diagnosis.

Although the precise time of sepsis treatment is debatable, it is generally agreed that early diagnosis and treatment are essential for increasing sepsis survival rates and lowering sepsis-related expenses [3,7]. Each hour of delayed treatment has been associated with roughly a 4–8% increase in mortality [3], further underscoring the importance of timely recognition and initiation of treatment. The lack of reliable blood- or plasma-based biomarkers makes early diagnosis even more difficult. Despite considerable efforts to identify biomarkers as sepsis prognostic indicators, none have proven to be sensitive or specific enough to be utilized commonly in clinical practice [8]. The available scoring systems [9–11], although useful for predicting general deterioration or mortality, cannot identify sepsis with high sensitivity and specificity at an individual level. Hence, there is a fundamental, unmet need for a clinical decision system capable of accurate and timely sepsis diagnosis.

The increased use of electronic health records, which can be queried in real-time, has presented opportunities to use Machine Learning (ML) approaches to identify patients at risk for sepsis [12–14]. However, previous studies have either used the older definition of sepsis or are designed in a way that makes integration into routine critical care challenging or have low

performance in detecting sepsis [15]. The low specificity of the current prediction algorithms and scoring systems leads to the occurrence of false alarms that may result in alert fatigue among healthcare professionals, thereby reducing their sensitivity and confidence in the algorithm. Failure to detect false alarms may lead to misdiagnosis, resulting in poor allocation of attention and hospital resources. Furthermore, it may lead to unnecessary prescription of antibiotics, leading to antibiotic resistance which is a growing concern in healthcare [16,17]. However, alarm fatigue can be addressed to an extent by maintaining a low false alarm rate and a high positive predictive value.

Similar work done by Persson et al. using electronic health record data has reported a high AUROC (Area under the Receiver Operating Characteristic Curve) value of 0.9 but reports a poor AUPRC (area under the precision-recall curve) value of 0.62 [18]. Focusing on the performance of AUROC alone can lead to unreliable models as it is less informative if the classes are highly imbalanced [19,20], which is often observed in sepsis studies [21–23]. Recently, a few studies also reported the use of clinician notes as input for training models along with other medical data [24,25], however, incorporating textual notes comes with its own challenges [26]. Few sepsis algorithms have been rigorously reviewed in terms of patient outcomes [27,28]. Existing algorithms using electronic health records (EHRs), such as EPIC, have a low positive predictive value and hence resulting in high false alarms [29]. The shortcomings of the present methodologies for sepsis prediction in the ICU and recommendations to address them are outlined in a systematic review by Moor et al. [15]. In this study, we developed a "*SepsisAI*" algorithm for predicting hospital-acquired sepsis in ICU patients six hours before the onset as per the sepsis-3 definition [2] by using clinical data gathered from electronic medical records. In addition, extra emphasis was given to addressing the critical issue of alert fatigue by significantly reducing the false-alarms ratio.

## Materials and methods

### Datasets

This study utilized a dataset obtained from the PhysioNet/Computing in Cardiology Challenge 2019 [30,31]. The dataset consisted of 40,336 de-identified ICU patient records gathered from Beth Israel Deaconess Medical Center, USA (n = 20,336) and Emory University Hospital, USA (n = 20,000). Patient characteristics are mentioned in the S1 Table. A combined total of 2,932 (7.26%) patients were tagged as sepsis positive in accordance with Sepsis-3 clinical criteria [2]. The data consisted of a combination of hourly vital sign summaries (n = 8), lab values (n = 26), and static patient descriptions (n = 6) (S1 Table). Patients with less than 8 hours of ICU data and sepsis onset occurring less than 4 hours after ICU admission were excluded from the dataset. The data was truncated after ICU discharge or after 2 weeks of ICU admission. A held-out dataset with 15% of random patients was kept aside for testing. The remaining dataset was randomly split at the patient level in the ratio of 80% and 20% for training and validation respectively.

### Sepsis labels

The labels indicating the sepsis status were available as a binary vector for each hour of the ICU stay. For sepsis negative patients, the labels for all the hourly data were always zero. For sepsis positive patients, the label before the onset of sepsis were labelled as zero, and after the onset of sepsis were labelled as one. For early prediction, the labels of sepsis onset were shifted up by six hours from actual onset to obtain the six-hour lookahead time (S1 Fig) [30].

## Data preprocessing

Missing data can introduce bias and uncertainty in predictions, thus compromising the clinical efficacy of the outcome. To address this challenge, a forward-filling imputation strategy using a time-limited, parameter-specific, sample-and-hold approach [32] was employed. Specifically, for each parameter, the last available value was carried forward (sample-and-hold) for a limited time based on the nature of the parameter. The time interval between consecutive readings of a parameter was analyzed to determine the hold limits of the parameter. For the remaining missing values and completely absent parameters, -1 was used for substitution.

Parameter values were normalized in a common range using min-max normalization to prevent any potential bias and boost the convergence rate. The min-max limits for the parameters were established based on the interquartile ranges and inputs from clinicians. The minimum and maximum limits used for the normalization are shown in S1 Table. Any parameter value that falls outside of the designated range was truncated to the nearest extreme value. The data were then rescaled to the range of [1,5] to detect even minute changes in the parameter values. The values were transformed using the following formula:

$$
\mathcal{Y} = \begin{cases} -1, & \text{if } x \text{ is missing} \\ \dfrac{4(x - x_{\min})}{x_{\max} - x_{\min}} + 1, & \text{otherwise} \end{cases}
$$

## Feature selection

Parameters that were missing in more than 80% of the patients were removed from the analysis (S2 Fig). The remaining parameters were selected by considering their relevance, inclusion in sepsis scoring systems, and ensuring the presence of at least one representative parameter for vital organs such as respiration, liver, kidney, coagulation, and heart. Additionally, some handcrafted features were calculated from existing features. Shock index (ratio of heart rate and systolic blood pressure), the ratio of blood urea nitrogen and creatinine, the Modified Early Warning System (MEWS) score, and a partial Sequential Organ Failure Assessment (pSOFA) score were derived and are henceforth referred to as derived parameters. pSOFA was calculated using only the available limited SOFA score parameters (MAP-without drug information, Bilirubin, and Platelets) and hence termed as partial SOFA, a derivative of SOFA score. Subsequently, a final set of 21 parameters was used to train and test the *SepsisAI* algorithm. This set included 7 vital parameters, 8 lab parameters, and 2 demographic parameters, in addition to 4 derived parameters.

## Model input and classification

To process the time series data for each patient, a discretization approach was employed, transforming the data into overlapping matrices of size **n** x **m**, where **n** represents the number of hours and **m** represents the number of features. To address the substantial class imbalance in the dataset, characterized by only 7.2% of patients exhibiting sepsis, under sampling was done over the matrices from the entire sepsis-negative training cohort. This approach introduced increased heterogeneity within the negative data subset. Finally, the classifier was fed a matrix of dimensions 4x21 as input for further analysis.

Long Short-Term Memory network (LSTM), an attention-based neural network capable of handling long-term dependencies was used as a classifier for this study. Using a series of 'gates', each with its own RNN, the LSTM manages to keep, forget, or ignore data points based

on a probabilistic model. LSTMs also help solve the exploding and vanishing gradient problems. A masking layer was added to prevent missing values from participating in training. A normalization layer was added to stabilize the hidden state dynamics of the network. The overall architecture is shown in S3 Fig. Adam optimizer was used for learning rate optimization, ReLU activation was used to introduce non-linearity into the model, and binary cross entropy was used as the loss function.

## Warning and alert system

In order to reduce the false alarm ratio, the concept of 'warnings' and 'alerts' was introduced. A warning is issued every time the sepsis probability score from the model crosses the default threshold of 0.5. The alarm sets off only when 'x'–the number of warnings is issued within a prespecified window of 'w' hours. The parameters x and w can be modulated according to the clinicians' preferences regarding sensitivity and specificity.

## Evaluation criteria

The algorithm was evaluated on two held-out test sets. The first held-out test set (n = 864) comprised an equal representation of samples from both the negative (n = 432) and positive (n = 432) classes. The evaluation was done at both the "time level" and the "patient level". The model's performance at the time level was evaluated by comparing the sepsis probability scores for each hour with the sepsis labels. The performance at the patient level was evaluated holistically by comparing the occurrence of alert with the presence of sepsis onset in the patient. In addition, a second held-out test set, which is a subset of the first held-out set, was constructed as per the prevalence of sepsis observed in our dataset (7.2%) resulting in 34 positive and 432 negative samples. The model was evaluated using metrics sensitivity, specificity, accuracy, AUROC, AUPRC, F1 scores, and False alarm ratio. The false alarm ratio was also calculated using the formula:

$$False\ Alarm\ Ratio = (FP/(TP+FP)) \times 100.$$

Different studies use various evaluation criteria, and these criteria don't always represent the practical utility of sepsis diagnosis and treatment. Hence, a utility score [30] was used as an evaluation metric to evaluate the timeliness along with the accuracy of the prediction. The scoring function rewards the algorithm for early predictions and penalizes the algorithm for late or missed predictions and for false alerts (S1 Text). The utility score attains a maximum value of 1 when all the sepsis positive patients are identified exactly at 6-hours before the actual onset with no false positives.

## Results

### Time-limited sample and hold approach for missing value imputation

An analysis of the data quality for completeness revealed that a substantial proportion (79%) of input data points were missing. To determine the hold limits for imputing the missing values, an examination of the time interval between consecutive readings for each parameter was conducted. It was observed that laboratory parameters were repeated within 24 hours and vital signs within 4 hours for at least 75% of the patients, including both sepsis and non-sepsis cases (S4 Fig). Accordingly, 4 hours and 24 hours were selected as the hold limits for vital and laboratory parameters, respectively. After the data imputation, it was observed that on an hourly basis, the missingness of the data reduced from 79% to 22% (S5 Fig).

### Model shows good performance in early prediction of sepsis

At the time-level, where the model's prediction on hourly data aggregated from all patients was compared to the corresponding sepsis status for that hour, the algorithm achieved an AUROC of 0.94 and AUPRC of 0.87 on the test dataset (Fig 1). The model performed well with accuracy, specificity, and sensitivity of 95.62%, 96.76%, and 85.40%, respectively (S6 Fig). Notably, the model also achieved a decent utility score of 0.77, demonstrating its ability to provide early predictions of sepsis. However, to accurately quantify the look-ahead time of predictions for individual patients and to comprehensively assess predictive performance

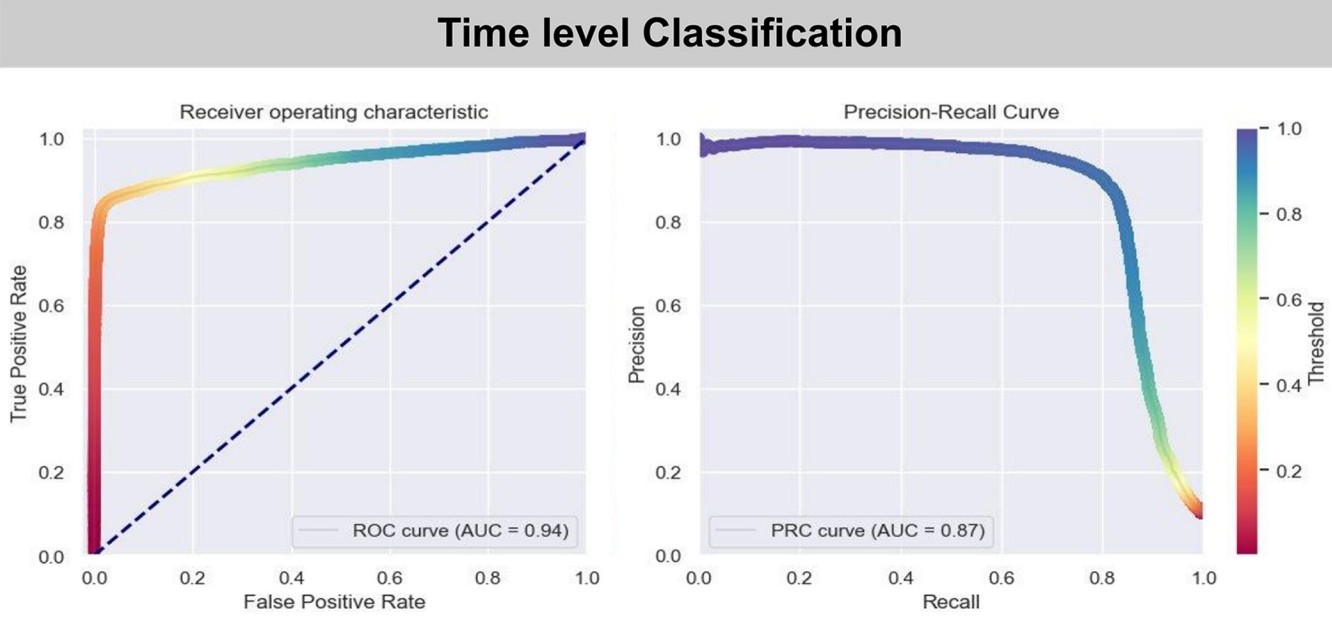

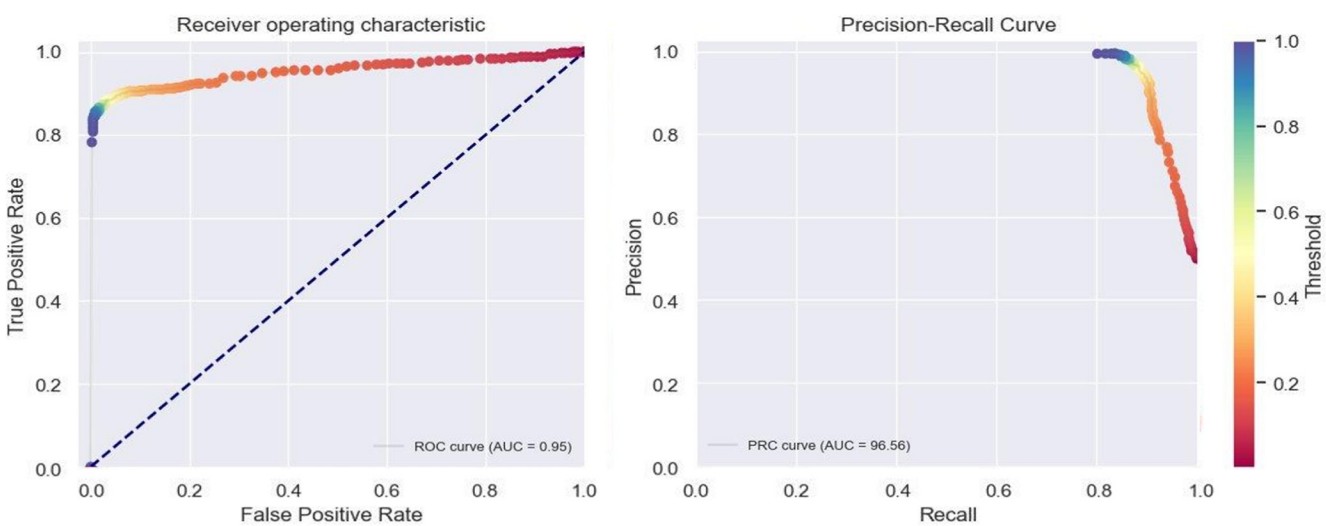

**Fig 1. Receiver Operating Characteristic and Precision-Recall curves for the time and patient level classification.**

**Table 1. SepsisAI's performance at the patient level for warnings and alerts.**

| Metric | Warnings | Alerts |
|--------|----------|--------|
| Accuracy | 86.57% | 92.47% |
| Specificity | 80.09% | 96.75% |
| Sensitivity | 93.05% | 88.19% |
| F1-score | 87.39% | 92.14% |
| PPV | 82.37% | 96.45% |
| NPV | 92.02% | 89.12% |
| False Alarm Ratio | 21.39% | 3.18% |

considering the time-series aspect of the data, it is imperative to evaluate the results at the patient-level.

## Warnings and alerts significantly reduces the false-alarms at patient-level

At the patient level, the model continuously monitors the hourly probability predictions to raise warnings and alerts. Warnings are triggered every time the probability breaches the pre-defined threshold of 0.5. On evaluating the model's performance at the patient level solely based on warnings, an accuracy of 86.57% was achieved (Table 1). However, this approach resulted in a very high false-alarm ratio of 21.39%. To control the false alarms, a method was implemented wherein alerts were raised based on the number of warnings 'x' within a specific time window 'w' (Fig 2).

Experimentation with various combinations of 'x' and 'w' demonstrated that larger window sizes 'w' had an adverse effect on sensitivity, resulting in missed detections. Conversely, as the threshold for the number of warnings 'x' within the window 'w' increased, specificity improved at the expense of sensitivity (Fig 3A). Optimum performance with respect to specificity and sensitivity was achieved when three warnings were issued within a 5-hour window for raising an alert (Fig 3A). Hence, x = 3 and w = 5 were used for reporting patient-level performance, and with this

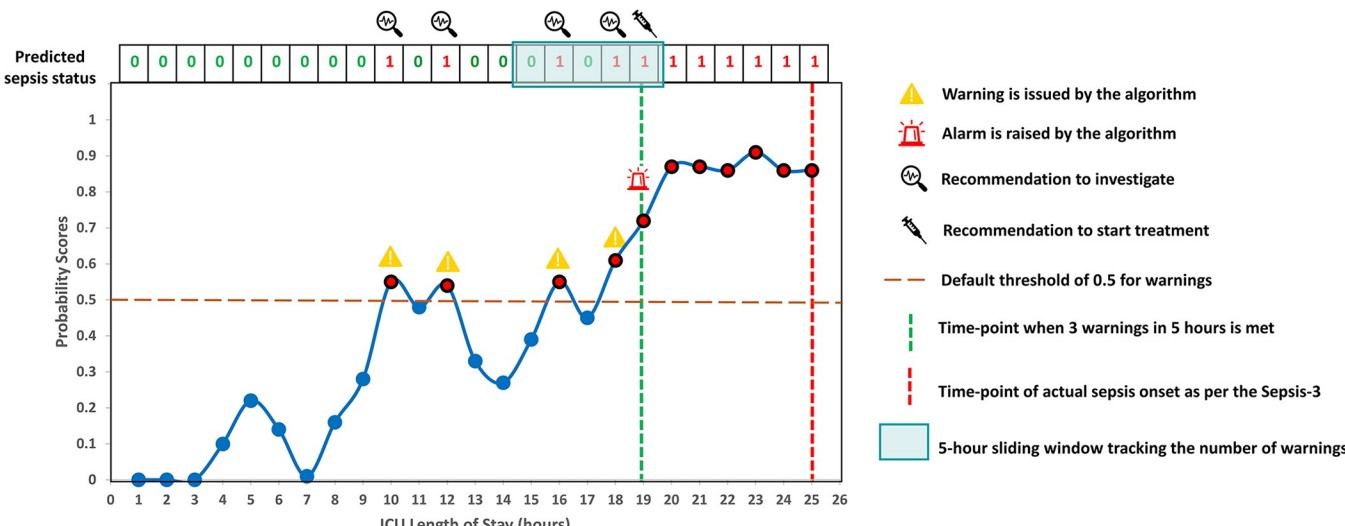

**Fig 2. An example of implemented 'Warning' and 'Alert' system and associated recommendations.** Warnings are triggered every time the probability breaches the predefined threshold of 0.5. Alerts are raised based on the number of warnings 'x' within a specific sliding time window 'w'. Recommendations are issued at different time-points based on warnings and alerts.

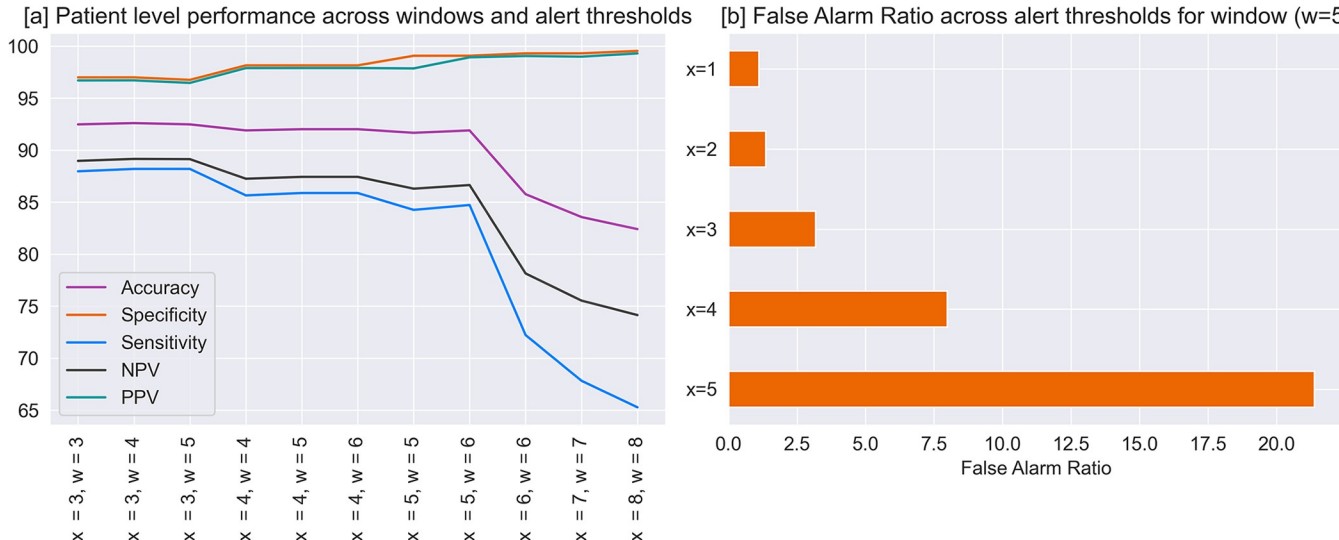

**Fig 3. Patient-level performance of the algorithm across different window sizes and alert thresholds.** [a] Effect of alert threshold 'x' and window size 'w' on the patient level performance. [b] Effect of modulating the alert threshold on the false alarm ratio keeping the window size constant.

configuration, the model exhibited patient-level AUROC, AUPRC, accuracy, sensitivity, and specificity, of 0.96, 0.97, 92.47%, 88.19%, and 96.75%, respectively (Table 1 and Fig 1). More importantly, with these regulations in place, the false-alarm ratio reduced significantly to 3.18% (Fig 3B). The algorithm also achieved exceptional performance on the second held-out test set, which was based on disease prevalence, with AURPOC, AUPRC, accuracy, specificity, and sensitivity, PPV, and NPV of 0.94, 0.87, 96.78%, 96.75%, 97.05%, 70.21%, and 99.76% respectively.

## Algorithm provided timely predictions for starting early treatment

Consistent with the primary objective of the algorithm, which is to provide timely predictions to offer a lookahead time for clinicians to initiate early treatment, the model successfully triggered the first warning at a median of 6 hours (interquartile range: 6 to 20 hours; Fig 4A, S7 Fig) and raised an alert at a median of 4 hours (interquartile range: 2 to 5 hours; Fig 4B, S7 Fig). The distribution of lookahead time was further investigated and it was observed that for more than 50% of the patient's alert was raised at least 4 hours before the actual onset and for 8.0% of patients the alarm was raised at least 10 hours before the actual onset (Fig 4C). As the lookahead time increased the prediction further in advance becomes more challenging. Nonetheless, these numbers underscore the timeliness of the algorithm's predictions, which is crucial in the effective treatment of sepsis.

Given the clinical distinction between patients with sepsis onset upon ICU admission and those who develop sepsis later, with the former cases already suspected of sepsis, an additional analysis was performed to evaluate algorithm's performance in both cases. The analysis showed that the performance remains consistent regardless of sepsis onset timing relative to ICU admission (S5 Table), highlighting its reliability in detecting and predicting sepsis cases within ICU setting.

## Clinically actionable recommendations based on warnings and alerts

Despite the relatively high false-alarm rate for warnings, it is noteworthy that in 88.19% of sepsis-positive patients, a warning was subsequently followed by a true alert. This observation

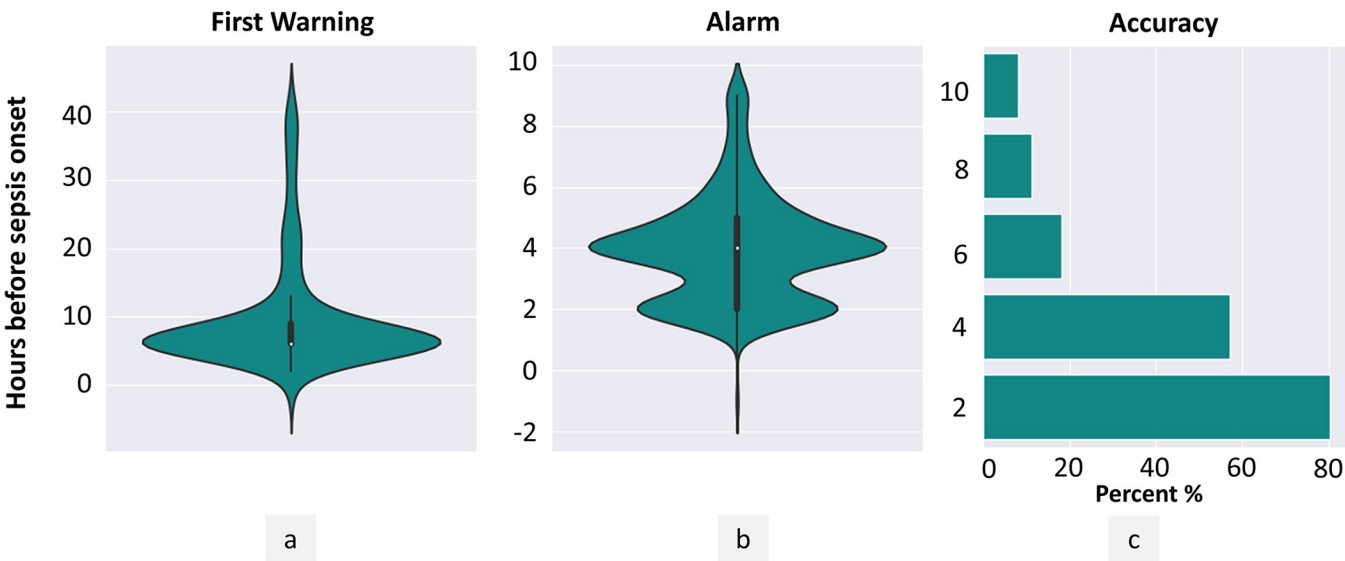

**Fig 4.** [a] A violin plot depicting the achieved lookahead for triggering a warning before the sepsis onset [b] A violin plot depicting the achieved lookahead for raising an alert before the sepsis onset. [c] The performance of the algorithm across multiple lookahead times in terms of accuracy.

emphasizes the importance of both warnings and alerts in the context of sepsis prediction. Therefore, it is recommended that both warnings and alerts are given due importance in clinical decision-making. Based on the results, warnings and alerts can provide actionable insights to clinicians in the management of sepsis cases. Warnings serve as an initial indication, prompting clinicians to closely monitor the patient's condition for the next few hours and, if needed, conduct further investigations to assess the likelihood of sepsis. On the other hand, an alert indicates a high probability of sepsis, warranting confident triaging and prompt intervention by initiating the treatment as per the defined guidelines [33].

## SepsisAI reports significantly low false alarm rate as compared to other scoring systems

A comparative assessment of *SepsisAI* with alternative scoring systems demonstrated *SepsisAI*'s superior performance in terms of sensitivity, specificity, precision, recall, F1, and false alarm ratio (Fig 5). Although the SIRS exhibits relatively high sensitivity, it lacks specificity, which results in a high rate of false alarms. Conversely, pSOFA and National Early Warning Score (NEWS) exhibit inferior performance in comparison. It is noteworthy, that *SepsisAI* achieved a false-alarm ratio of ~3% which is significantly less than other sepsis alarm systems such as SIRS (41.35%), NEWS (43.26%), and pSOFA (35.33%).

## The individual patient profile gives deeper insights into patient's risk profiles

The hourly risk scores from the model were plotted to better understand the patients' risk profiles. An in-depth analysis of these risk profiles provided valuable insights into the impact of warnings and alerts. Fig 6A portrays the ideal risk profile of a patient without sepsis, wherein the probability scores consistently remain below the threshold of 0.5. Consequently, the algorithm classifies the patient as sepsis negative. In certain instances of sepsis-negative cases (Fig 6B), the probability threshold was breached, triggering warnings from the algorithm. However, these warnings were isolated, failing to satisfy the alert criterion of three warnings

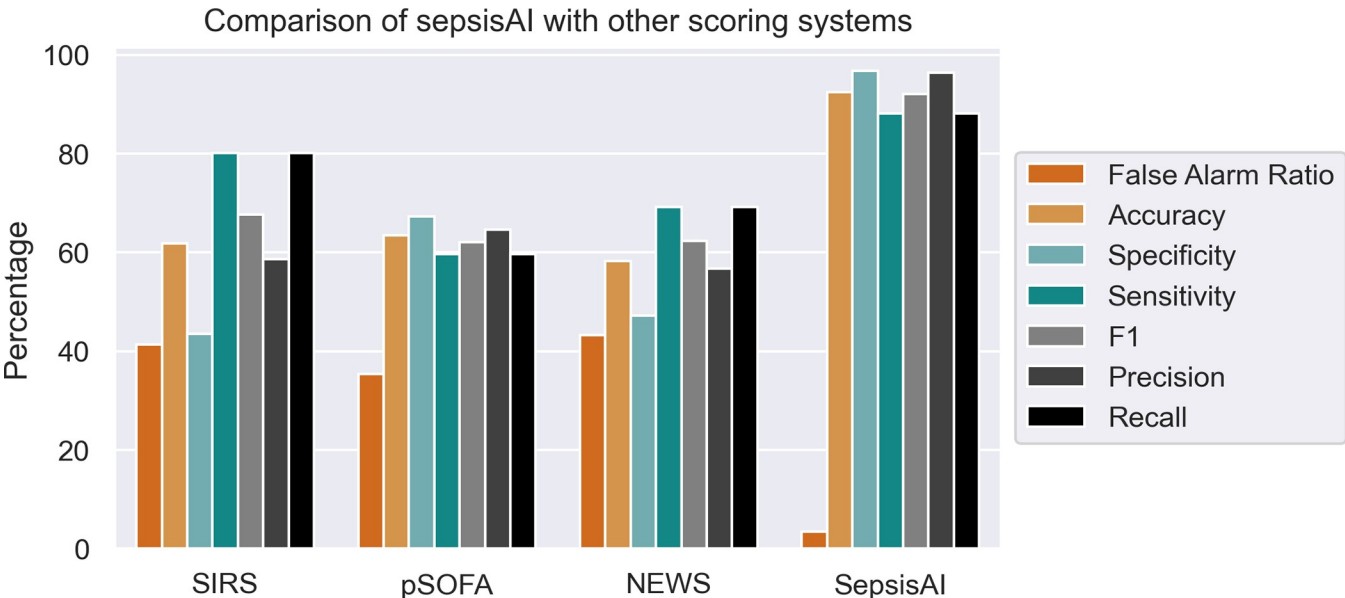

**Fig 5. Grouped bar plot depicting the performance comparison of SepsisAI with other scoring systems. SIRS- Systematic inflammation response syndrome, pSOFA–partial sequential organ failure assessment, NEWS–National early warning score.**

within a 5-hour window. As a result, false alarms for sepsis-negative patients were effectively avoided. This example perfectly illustrates the utility of the alert system in reducing the false alarms. In the case of sepsis-positive patients, Fig 6C exhibits the ideal risk score profile of a patient with sepsis, wherein the criterion of three warnings within a five-hour window was met, raising a true alarm 10 hours prior to the actual onset. This valuable lookahead time enables clinicians to initiate treatment in advance. Fig 6D depicts an interesting scenario where few warnings were triggered before raising the alarm. Although the initial warnings could not meet the alert criteria, however within 4 hours the next set of warnings was issued which resulted in raising a true alarm. Such scenarios bolster our recommendation that warnings should prompt close monitoring of the patient for the next few hours. A comprehensive version of these figures along with the parameters is shown in the S8 Fig.

## Discussion

Timely recognition of sepsis is crucial in facilitating prompt initiation of sepsis bundles, effectively preventing disease progression. The development of accurate early sepsis prediction models, therefore, hold immense potential to significantly improve patient outcomes and alleviate the burden of sepsis on the healthcare system. However, defining sepsis retrospectively is complex, as it encompasses multiple elements such as culture orders, antibiotic administration, and changes in SOFA scores in line with the Sepsis-3 definition.

To contextualize the results of this study, it's essential to understand what the algorithm predicts. Based on the sepsis definition provided, it can be inferred that the algorithm doesn't predict sepsis-related events in isolation. Instead, it predicts organ failure within the context of infection by continuously monitoring representative markers of infection (e.g., WBC, temperature, HR) and organ function (e.g., creatinine, BUN for kidneys, bilirubin for the liver). This is supported by the observation that the negative cohort includes patients with either organ failure or suspected infection. The algorithm's performance indicates that it predicts sepsis only for patients meeting both criteria.

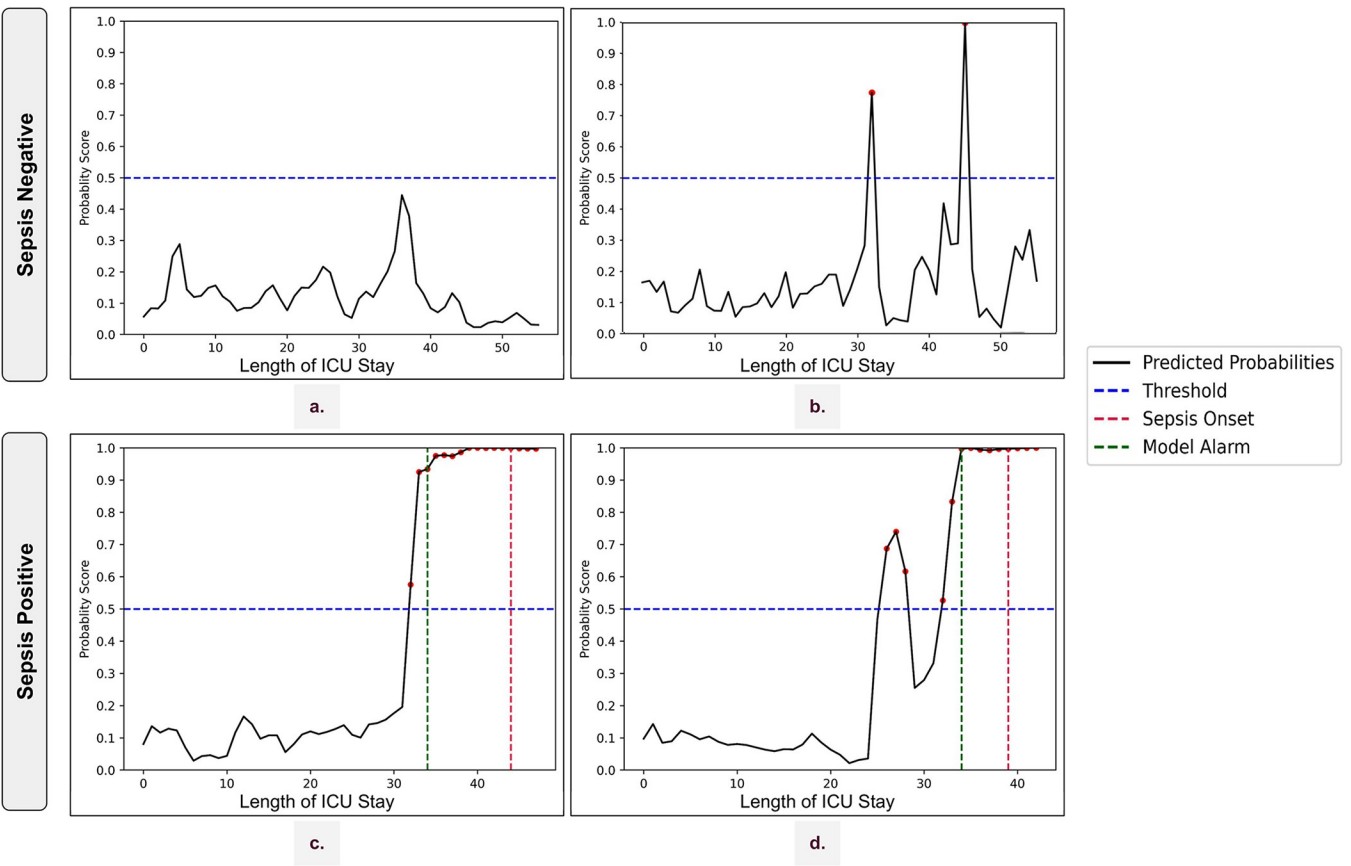

**Fig 6.** Individual risk profiles for [a and b] sepsis-negative and [c and d] positive patients. The probability of sepsis (y-axis) is plotted against the length of stay (x-axis) of the patient. A warning is issued if the risk is greater than 0.5 (blue horizontal line). The alarm is raised once there are 3 warnings in a 5-hour window (green vertical line). The actual sepsis onset is denoted using the red vertical line.

The *SepsisAI* algorithm is an important first step toward the clinical adoption of an ML approach for sepsis prediction. This algorithm addresses two key issues–alarm fatigue caused by false alarms and the need for early prediction.

The integration of the warning and alert system into our algorithm has proven instrumental in significantly reducing false alarms. While the primary objective of the alert criteria is to minimize false positives resulting from isolated warnings, it has been observed that in 88.19% of sepsis-positive patients, a warning is followed by a true alarm. Therefore, it is advised that clinicians closely monitor patients for the next few hours who receive a warning and conduct further investigations, if needed, to triage them. However, as soon as the alert criteria is met, given the algorithm's high accuracy and low false alarm ratio, it is recommended for the clinician to promptly attend to the patient and initiate sepsis bundles which might involve antibiotic administration, fluid resuscitation, etc. We anticipate that the clinicians will be able to make well-informed decisions on sepsis patient management by combining the *SepsisAI*'s output with other clinical information, such as patient history, signs and symptoms, and risk factors.

From the clinical significance perspective, it is important to know the expected disease prevalence in the target population to mitigate the potential harms and costs of the predictive model. Disease prevalence impacts the positive and negative predictive values of a predictive model, influencing subsequent decisions and interventions. Validating the algorithm on a test

set representing the actual prevalence of sepsis confirmed its high performance and utility. It is worth noting that the prevalence of sepsis may vary across different healthcare systems. Hence, the flexibility to adjust the window size w and alert threshold x enables clinicians to modulate the specificity and sensitivity of the alerts to align with their specific requirements.

The timing of alerts is a critical consideration to avoid raising alerts too early, which may be disregarded by the clinicians, or too late, which may not allow sufficient time for appropriate action. *SepsisAI* raises a warning at a median of six hours and an alert at a median of four hours before the sepsis onset, respectively, thus providing an optimal time for timely clinical interventions. This is crucial considering the documented increase in mortality by 4–8% with each hour of delayed antibiotic administration [3].

While *SepsisAI* algorithm has shown promising results, this study has some limitations. There needs to be further validation on bigger cohorts from other hospitals across different types of ICUs with enough representation of populations with comorbidities. Our results cannot be claimed to be generalized beyond the scope of the patient population under consideration which, while geographically and institutionally heterogeneous, does not represent all patient populations of interest. There needs to be further validation of external datasets and prospective studies. Additionally, due to the variations in the training and testing data, set of input parameters, and available models or complete implementation details, it is not feasible to compare our results with the existing algorithms. The comparator scoring systems, such as pSOFA, SIRS, and NEWS, may not fully capture the complexity of sepsis detection in the ICU setting. However, given the absence of standardized criteria for sepsis detection and the lack of implementable comparable research, we chose to utilize these commonly employed scoring systems as comparators in our study. pSOFA is derived from some of the parameters used to determine the sepsis onset and hence may introduce some degree of circularity.

Our ongoing efforts are dedicated to analyzing the true impact of such an algorithm, as we seek to validate our approach further and deploy it prospectively within the hospital setting. Future work may address further practical challenges hindering the clinical adoption of ML-based early warning systems. Additionally, the algorithm could be customized to a particular site by leveraging data from other hospitals. Another key consideration in clinical adoption is the provision of clear and comprehensive explanations of the system's decisions to the clinicians to foster trust and accountability and mitigate any biases [34]. Explainability methods such as SHAP [35], Integrated Gradients [36] and LIME [37] have been proposed as means of achieving this aim. However, these techniques suffer from inherent limitations, making it difficult to interpret their results, particularly in complex time series and multivariate analyses.

## Conclusion

In this retrospective study, an algorithm is proposed to serve as a clinical decision support system assisting clinicians in the accurate and timely diagnosis of sepsis. With exceptionally high specificity and low false-alarm ratio, this algorithm also helps mitigate the well-known problem of clinician alert fatigue that arises from currently implemented sepsis alert systems. Consequently, the algorithm partially addresses the challenges of successfully integrating machine-learning algorithms into routine clinical care. We hope it will ultimately improve patient survival and relevant antimicrobial stewardship outcomes for heterogeneous and complex s such as sepsis. Randomized clinical trials are required to further establish the model's usefulness by comparing sepsis management based on its use vs. standard identification of sepsis with respect to clinically relevant primary endpoint such as short-term mortality.

## Supporting information

**S1 Text. Utility Score.**
(DOCX)

**S1 Fig. Sepsis Onset Tagging.**
(DOCX)

**S2 Fig. Missingness of parameters in patients.**
(DOCX)

**S3 Fig. Model Architecture.**
(DOCX)

**S4 Fig. Time difference between two consecutive measurements of [a] lab parameters and [b] vital signs.**
(DOCX)

**S5 Fig. Bar chart showing the parameter availability before and after imputation across selected parameters. The patients with a parameter completely absent were excluded from the analysis.**
(DOCX)

**S6 Fig. Confusion matrix for alerts at the time and patient level on a balanced test set.**
(DOCX)

**S7 Fig. Histogram depicting the number of hours before onset when the first warning was issued, and the first alert was raised.**
(DOCX)

**S8 Fig. Parameter values across time.**
(DOCX)

**S1 Table. Patient Characteristics.**
(DOCX)

**S2 Table. List of available parameters and their descriptions with units.**
(DOCX)

**S3 Table. Minimum and maximum values used for normalizing parameters.**
(DOCX)

**S4 Table. Model performance at the time and patient level considering alerts.**
(DOCX)

**S5 Table. Model performance (patient level) w.r.t sepsis onset from ICU admission.**
(DOCX)

## Acknowledgments

The authors would like to acknowledge Dr. Donald Chalfin (Siemens Healthineers) for the clinical inputs and Dr. Megan Carpenter (Siemens Healthineers) for reviewing the manuscript.

## Author Contributions

**Conceptualization:** Ankit Gupta.

**Formal analysis:** Ankit Gupta, Ruchi Chauhan, Saravanan G, Ananth Shreekumar.

**Investigation:** Ankit Gupta.

**Methodology:** Ankit Gupta, Saravanan G, Ananth Shreekumar.

**Project administration:** Ankit Gupta.

**Software:** Saravanan G, Ananth Shreekumar.

**Supervision:** Ankit Gupta.

**Validation:** Ruchi Chauhan, Saravanan G.

**Visualization:** Ankit Gupta, Ruchi Chauhan.

**Writing – original draft:** Ankit Gupta, Ruchi Chauhan.

**Writing – review & editing:** Ankit Gupta, Ruchi Chauhan, Saravanan G, Ananth Shreekumar.

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
