## [Decision Letter · Decision Letter 0]

6 Feb 2024

PDIG-D-23-00409

Improving Sepsis Prediction in Intensive Care with SepsisAI: A clinical decision support system with a focus on minimizing false alarms

PLOS Digital Health

Dear Dr. Gupta,

Thank you for submitting your manuscript to PLOS Digital Health. After careful consideration, we feel that it has merit but does not fully meet PLOS Digital Health's publication criteria as it currently stands. Therefore, we invite you to submit a revised version of the manuscript that addresses the points raised during the review process.

Please submit your revised manuscript within 60 days Apr 06 2024 11:59PM. If you will need more time than this to complete your revisions, please reply to this message or contact the journal office at digitalhealth@plos.org. Please include the following items when submitting your revised manuscript:

We look forward to receiving your revised manuscript.

Kind regards,

Fei Wang

Guest Editor

PLOS Digital Health

Journal Requirements:

Additional Editor Comments (if provided):

Both reviewers have raised concerns. Please address their comments and send back a revised version with point-by-point responses.

Reviewers' comments:

Reviewer's Responses to Questions

**Comments to the Author**

1. Does this manuscript meet PLOS Digital Health’s publication criteria? Is the manuscript technically sound, and do the data support the conclusions? The manuscript must describe methodologically and ethically rigorous research with conclusions that are appropriately drawn based on the data presented.

Reviewer #1: Partly

Reviewer #2: Yes

2. Has the statistical analysis been performed appropriately and rigorously?

Reviewer #1: Yes

Reviewer #2: I don't know

3. Have the authors made all data underlying the findings in their manuscript fully available (please refer to the Data Availability Statement at the start of the manuscript PDF file)?

Reviewer #1: Yes

Reviewer #2: Yes

4. Is the manuscript presented in an intelligible fashion and written in standard English?

Reviewer #1: Yes

Reviewer #2: Yes

5. Review Comments to the Author

Reviewer #1: This is a moderately sized, retrospective, ICU, sepsis-3 machine learning prediction time series analysis. There is inherent value in predicting severe infection and there is theoretical potential benefit in earlier treatment and management. Strengths of this analysis include the detailed time windowing of the prediction period and the accuracy. The analysis of the trade off’s between windowing, and number of alerts is useful. Weaknesses include the limitation to an ICU cohort, that by definition are already closely monitored, and lack of specifics about the components of Sepsis-3 that are being predicted. 

Major: 

1. Central to the derivation of benefit from prediction is a deep understanding of the disparate components of the current epidemiologic definition of sepsis, Sepsis-3. This definition is a combination of cultures (for diagnosis), antibiotics (treatment) and an increase or absolute SOFA score > 2 (a presumed effect of the infection). Each of these components has a different biologic and sociologic meaning. Specifically, the increase in SOFA score may be the result of a severe infection but could also be caused by mechanical (such as a pulmonary embolism or bleeding) or iatrogenic complications. Moreover, the decisions to attempt to diagnosis or treat a potential infection are reflective of the sociology of the unit under consideration. Thus, predicting sepsis under this construct is NOT PREDICTING an infection. It is predicting several things that are tangentially related to an infection. That said, the physionet challenge was to predict sepsis in this construct, so it is not the fault of the authors. For the results to be put into context, it must be clear as to what is being predicted. As these events occur in series, the final component finishes the definition. Is the prediction antibiotic ordering, a change in SOFA, or the ordering of cultures? It is most important to predict the active orders as this may lead to earlier treatment, however if the majority of cases are predicting a change in SOFA, when antibiotics are already in place then the prediction has less value. 

2. The results pertaining to the look ahead period, the warning v alert accuracy etc need to be referenced to the above concern. If the insights are to be actionable, then the components predicted must be clear. 

3. The selected derived features include the outcome. More specifically, the pSOFA is part of the outcome definition of sepsis 3. Please justify this selection. 

4. The comparator scoring systems are not designed to predict in ICU sepsis. SIRS, and NEWS were designed as screening concepts for patients outside of the ICU. The SOFA score is not a prediction system, it is multiple organ dysfunction score and part of the definition of sepsis, without the concern for infection. This speaks to the relatively selected population for this exercise, see comment 6 below. 

5. It may be useful to contextualize the results of this exercise in terms of the eventual outcomes of the predicted sepsis cases v those without the event. Specifically, what are the outcomes for those with predicted sepsis v without predicted sepsis. How many had a documented infection? How many continued antibiotics for more than the initial dose? How many survived the ICU stay? 

6. The framing of the manuscript speaks of general sepsis prediction; however this is a selected ICU cohort. Sepsis is one of the most common causes of admission to an ICU, so selecting ICU onset sepsis (meaning not present on admission) creates bias that may limit the generalizability of any findings. This needs to be discussed and put into context. Moreover, why was the 4 hour cut off selected for inclusion into the cohort? Sepsis 3 onset 6 hours after arrival to the ICU is much different than Sepsis onset on day 3 after arrival to the ICU. It may be argued that the 6 hour sepsis onset is simply a labeling problem, meaning that everyone knew that the patient had sepsis prior to criteria being met. The day 3 sepsis is potentially a new event. 

7. Figure 3 has some value but lacks clinical context. It would be useful to display the collected set of features, e.g. vital signs, antibiotic orders, and lab values etc for the referent patients in the panels. 

8. When were the sepsis events in this study relative to ICU admission? Was performance different in different epochs after ICU admission? 

9. What was the rationale to rescale with the utilized formula, line 139?

Reviewer #2: Thank you for the opportunity to review the manuscript titled “Improving sepsis prediction in intensive care with SepsisAI: a clinical decision support system with a focus on minimizing false alarms”. This paper presents a commendable effort in developing a machine learning algorithm to predict the onset of sepsis in ICU patients. The methodology involving a detailed evaluation of the model at both the "time level" and "patient level" is particularly noteworthy. The approach demonstrates a lower false alarm rate as compared to other scoring systems. However, I have a few specific comments and suggestion that could potentially enhance the quality and clarity of the research:

1) The paper would benefit from a more detailed description of the patient population demographics in the dataset. Understanding the age, gender, race, and other demographic factors is crucial for assessing the generalizability of the model.

2) The method used for splitting the data into training, validation, and test sets needs further clarification. It's unclear whether the data was randomly split or if there was another stratification process involved.

3) Regarding the second held-out test set, the number of positive cases (34) seems too small for a comprehensive evaluation of the model's performance. If feasible, expanding this to include 432 positive cases and 6000 negative cases could provide a more rigorous assessment of the model's predictive capabilities.

4) While the forward-filling imputation strategy to address missing values is a smart choice, there's room to reconsider the time frame for certain rapidly changing laboratory values in sepsis conditions, such as lactate, some CBC parameters, PT, or procalcitonin. Employing a 12-hour window instead of a 24-hour one might yield more accurate results, given that laboratory values can fluctuate significantly in a short period for ICU patients.

5) Evaluating the model in a prospective patient population would be helpful to understand its real-world utility. I am interested in knowing the next steps. Maybe the authors can discuss about it. 

Overall, the research presents a significant step forward in the application of machine learning in a critical care setting. Addressing these comments could further strengthen the findings and their implications in the clinical environment.

6. PLOS authors have the option to publish the peer review history of their article (what does this mean?). If published, this will include your full peer review and any attached files.

**Do you want your identity to be public for this peer review?** For information about this choice, including consent withdrawal, please see our Privacy Policy.

Reviewer #1: Yes: Edward James Schenck

Reviewer #2: No

---

## [Decision Letter · Decision Letter 1]

21 Jun 2024

PDIG-D-23-00409R1

Improving Sepsis Prediction in Intensive Care with SepsisAI: A clinical decision support system with a focus on minimizing false alarms

PLOS Digital Health

Dear Dr. Gupta,

Thank you for submitting your manuscript to PLOS Digital Health. After careful consideration, we feel that it has merit but does not fully meet PLOS Digital Health's publication criteria as it currently stands. Therefore, we invite you to submit a revised version of the manuscript that addresses the points raised during the review process.

Please submit your revised manuscript within 30 days Jul 21 2024 11:59PM. If you will need more time than this to complete your revisions, please reply to this message or contact the journal office at digitalhealth@plos.org. Please include the following items when submitting your revised manuscript:

We look forward to receiving your revised manuscript.

Kind regards,

Fei Wang

Guest Editor

PLOS Digital Health

Journal Requirements:

Additional Editor Comments (if provided):

Reviewers' comments:

Reviewer's Responses to Questions

**Comments to the Author**

1. If the authors have adequately addressed your comments raised in a previous round of review and you feel that this manuscript is now acceptable for publication, you may indicate that here to bypass the “Comments to the Author” section, enter your conflict of interest statement in the “Confidential to Editor” section, and submit your "Accept" recommendation.

Reviewer #1: (No Response)

Reviewer #2: All comments have been addressed

2. Does this manuscript meet PLOS Digital Health’s publication criteria? Is the manuscript technically sound, and do the data support the conclusions? The manuscript must describe methodologically and ethically rigorous research with conclusions that are appropriately drawn based on the data presented.

Reviewer #1: Yes

Reviewer #2: Yes

3. Has the statistical analysis been performed appropriately and rigorously?

Reviewer #1: Yes

Reviewer #2: Yes

4. Have the authors made all data underlying the findings in their manuscript fully available (please refer to the Data Availability Statement at the start of the manuscript PDF file)?

Reviewer #1: Yes

Reviewer #2: Yes

5. Is the manuscript presented in an intelligible fashion and written in standard English?

Reviewer #1: Yes

Reviewer #2: Yes

6. Review Comments to the Author

Reviewer #1: It is still not clear to me whether the sepsis challenge allows for the prediction of T suspicion of sepsis and T sofa and what these mean. The authors should strongly consider a sensitivity analysis to determine whether T suspicion can be predicted (this is an action by clinicians cultures and antibiotics) and may be more valuable. “Sepsis” onset may not be valuable without knowing the components. I understand that this may be limited by the available data of the sepsis data challenge. This is not the fault of the authors but the fault of the challenge organizers. 

pSOFA is adding circularity to the prediction model. Either remove it from the model or discuss it as a limitation

Reviewer #2: Thank you for addressing my questions. I think the manuscript is acceptable for publication.

7. PLOS authors have the option to publish the peer review history of their article (what does this mean?). If published, this will include your full peer review and any attached files.

**Do you want your identity to be public for this peer review?** For information about this choice, including consent withdrawal, please see our Privacy Policy. 

Reviewer #1: Yes: Edward Schenck

Reviewer #2: No

---

## [Editor Report · Decision Letter 2]

1 Jul 2024

Improving Sepsis Prediction in Intensive Care with SepsisAI: A clinical decision support system with a focus on minimizing false alarms

PDIG-D-23-00409R2

Dear Dr. Gupta,

We are pleased to inform you that your manuscript 'Improving Sepsis Prediction in Intensive Care with SepsisAI: A clinical decision support system with a focus on minimizing false alarms' has been provisionally accepted for publication in PLOS Digital Health.

Best regards,

Fei Wang

Guest Editor

PLOS Digital Health